# Economic impact of a machine learning-based strategy for preparation of blood products in brain tumor surgery

**Thara Tunthanathip👤\*, Sakchai Sae-heng, Thakul Oearsakul, Anukoon Kaewborisutsakul, Chin Taweesomboonyat**

Division of Neurosurgery, Department of Surgery, Faculty of Medicine, Prince of Songkla University, Songkhla, Thailand

\* tsus4@hotmail.com

**Data Availability Statement:** All relevant data are within the paper and its Supporting Information files.

## Abstract

### Background

Globally, blood donation has been disturbed due to the pandemic. Consequently, the optimization of preoperative blood preparation should be a point of concern. Machine learning (ML) is one of the modern approaches that have been applied by physicians to help decision-making. The main objective of this study was to identify the cost differences of the ML-based strategy compared with other strategies in preoperative blood products preparation. A secondary objective was to compare the effectiveness indexes of blood products preparation among strategies.

### Methods

The study utilized a retrospective cohort design conducted on brain tumor patients who had undergone surgery between January 2014 and December 2021. Overall data were divided into two cohorts. The first cohort was used for the development and deployment of the ML-based web application, while validation, comparison of the effectiveness indexes, and economic evaluation were performed using the second cohort. Therefore, the effectiveness indexes of blood preparation and cost difference were compared among the ML-based strategy, clinical trial-based strategy, and routine-based strategy.

### Results

Over a 2-year period, the crossmatch to transfusion (C/T) ratio, transfusion probability (Tp), and transfusion index (Ti) of the ML-based strategy were 1.10, 57.0%, and 1.62, respectively, while the routine-based strategy had a C/T ratio of 4.67%, Tp of 27.9%%, and Ti of 0.79. The overall costs of blood products preparation among the ML-based strategy, clinical trial-based strategy, and routine-based strategy were 30, 061.56$, 57,313.92$, and 136,292.94$, respectively. From the cost difference between the ML-based strategy and routine-based strategy, we observed cost savings of 92,519.97$ (67.88%) for the 2-year period.

**Funding:** A grant was provided by the Faculty of Medicine, Prince of Songkla University. (Grant no. 64-477-10-1). The funders had no role in study design, data collection, analysis, decision to publish, or preparation of the manuscript.

**Competing interests:** There are no conflicts of interest to declare.

## Conclusion

The ML-based strategy is one of the most effective strategies to balance the unnecessary workloads at blood banks and reduce the cost of unnecessary blood products preparation from low C/T ratio as well as high Tp and Ti. Further studies should be performed to confirm the generalizability and applicability of the ML-based strategy.

## Introduction

Brain tumor surgery is one type of operation that presents the risk of intraoperative transfusions such as meningioma, suprasellar tumors, cerebellopontine angle tumors, and skull-based procedures [1–3]. However, the literature review showed excessive blood products preparation has been observed in neurosurgical operations, which was estimated from the high crossmatch to transfusion (C/T) ratio. Chotisukarat et al. reported a C/T ratio for brain tumor operation at 5.0 [2], while Saringcarinkul et al. reported C/T ratios for meningioma, suprasellar tumor, and cerebellopontine angle tumor operations at 4.0, 4.2, and 8.7, respectively [3]. Over-ordering of preoperative blood products preparation can lead to an unnecessarily increased workload at blood banks as well as expired blood products. Hence, several management approaches have been studied to enhance the effectiveness of blood transfusions, such as patient blood management programs for the early detection and proper management of preoperative anemia [4], optimization of hemoglobin levels for transfusion [5], and preoperative autologous donation [6].

The pandemic caused by the Coronavirus (COVID-19) has negatively affected blood donations, which have decreased globally to approximately half that of typical periods [7–9]. Hence, the effectiveness of the protocol for preoperative blood products preparation should be a concern in situations involving limited resources. There are several strategies used to calculate the Maximum Surgical Blood Order Schedule (MSBOS) as follows: Procedure-based guideline [10, 11], 1.5 times the transfusion index [3, 7], and routine protocol [12].

Nowadays, machine learning (ML) has been studied in clinical research to support decision-making [13, 14]. However, how machine learning is used in general practice remains a challenge [15]. For example, an ML-based screening system for COVID-19 has been developed [16, 17] and subsequently implemented in clinical practice via mobile phone [18]. Additionally, Tunthanathip et al. used a random forest algorithm to optimize cranial computed tomography in children after a traumatic brain injury via a web application [19, 20].

The economic perspective is one of the outcomes that could be evaluated by ML implications because the over-requisition of preoperative blood products preparation has been reported in prior studies, which tends to increase the burdens on blood banks, as well as the wastage of blood products and unnecessary costs. Consequently, this research aimed to identify the cost differences of the ML-based strategy compared with other strategies in preoperative blood products preparation. Besides, a secondary objective was to compare the effectiveness indexes for blood products preparation among the strategies.

## Methods

### Study design and study population

The study utilized a retrospective cohort design conducted on patients who had been diagnosed with a brain tumor and undergone surgery between January 2014 and December 2021

at a single-center hospital. However, patients whose medical records contained incomplete transfusion data or who received a blood transfusion before surgery were excluded. Baseline clinical characteristics, preoperative hematologic laboratories, and operational data were collected from electronic-based medical records.

## Strategies for preoperative blood products preparation

In the present study, three strategies for preoperative blood products preparation were compared as follows: ML-based strategy, clinical trial-based strategy, and routine-based strategy.

## ML-based strategy

The first strategy for preoperative blood products preparation was developed by the training of the ML predictive model that forecasted the number of units of blood products from several datasets.

Total data in the present study comprised two major cohorts of data that corresponded with two processes of workflow as follows: 1) Development of predictive models and deployment of the ML-based web application, and 2) validation of the web application, as shown in Fig 1.

The first cohort involved data from 1,267 patients diagnosed with brain tumors between January 2014 and December 2019, which was divided into a training dataset and a testing dataset using the 70:30 random splitting method. Consequently, 887 patients were included in the training dataset, while the testing dataset comprised 380 patients for intraoperative blood transfusion predictability. Details concerning the development and deployment of the ML-based web application are described in the Supplement.

The second cohort comprised data from 414 patients who had undergone brain tumor surgery between January 2020 and December 2021. Data for this cohort was used for validation of the predictive models that were built via the web application, as shown in Fig 2. The tool is simple to use by scanning the quick response (QR) code or getting the uniform resource locator (URL) on laptops or smartphones, after which the web application will be ready to use. The

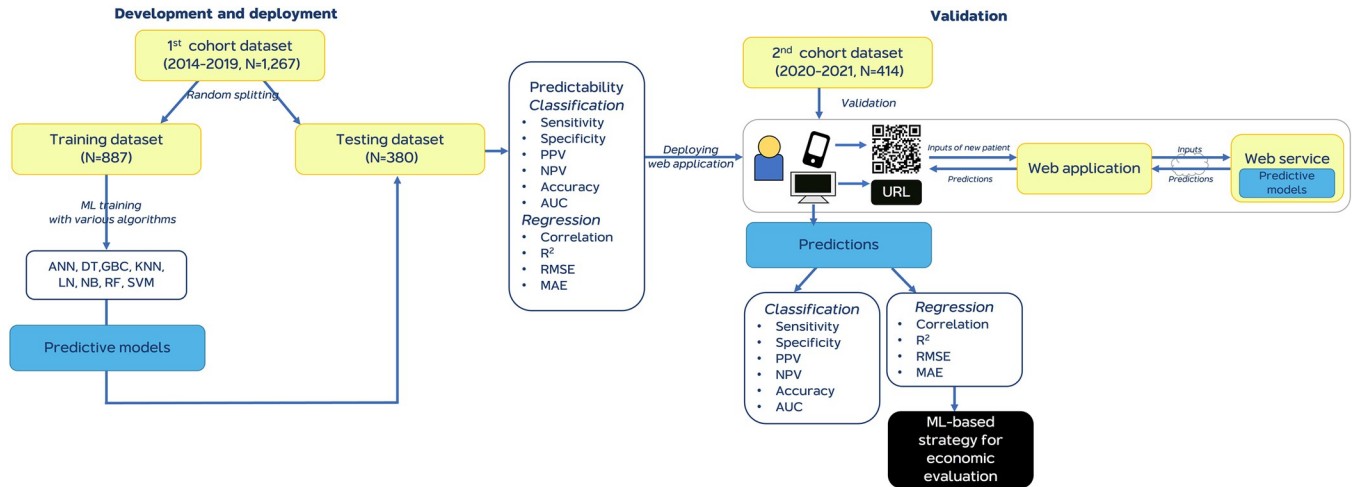

**Fig 1. Workflow of the ML-based strategy for preoperative blood preparation.** Abbreviations: ANN = artificial neural network, AUC = Area under the ROC curve, DT = Decision tree, GBC = gradient boosting classifier, MAE = Mean Absolute Error, ML = machine learning, NB = naïve Bayes, NPV = negative predictive value, PPV = positive predictive value, $R^2$ = R-squared, RF = random forest, RMSE = Root Mean Squared Error, SVM = support vector machine.

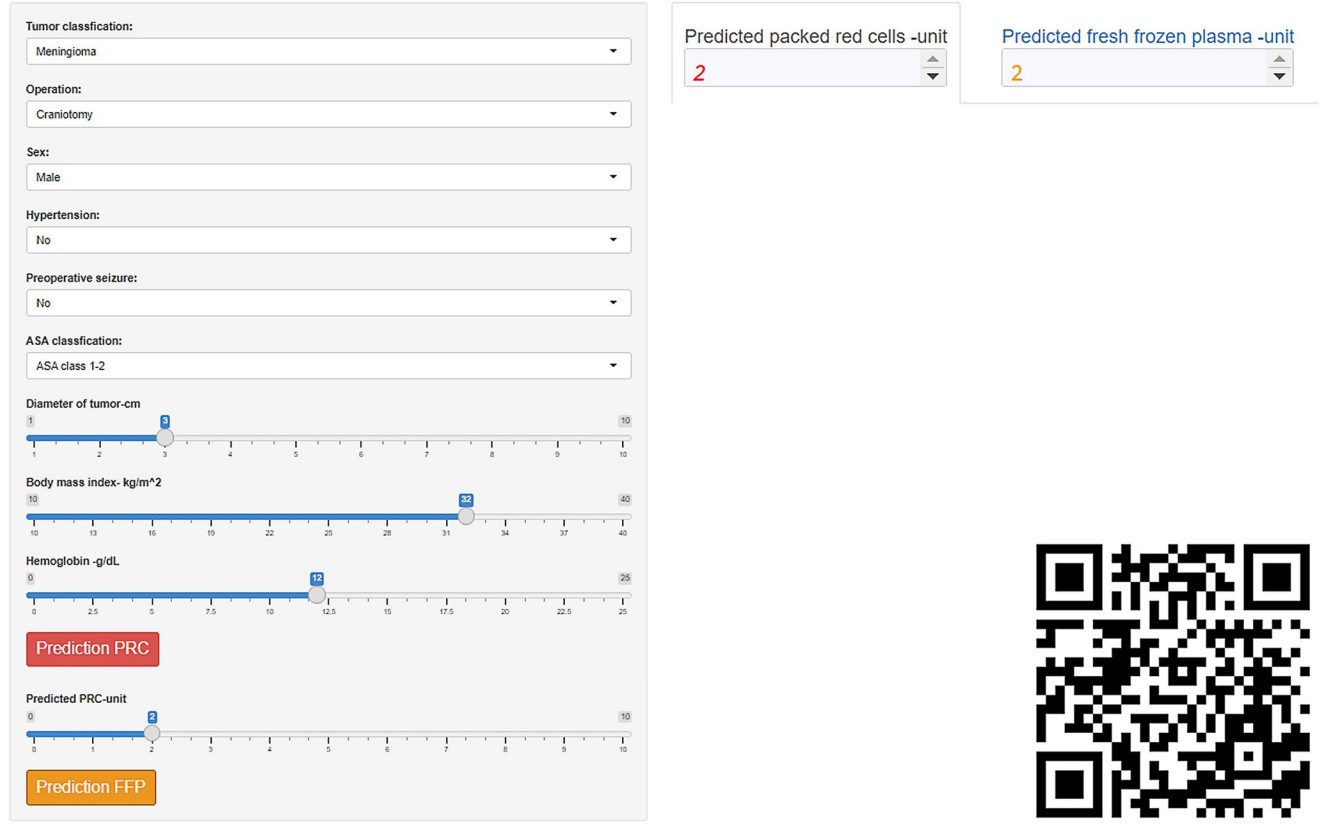

**Fig 2. Screenshot of the ML-based application for preoperative blood preparation.** The tool is used via QR code or https://neurosxpsu.shinyapps.io/crossmatch/. To use the web application, input the new patient's parameters and press the red bottom for the number of PRC units, then input the calculated PRC units and press the yellow bottom for the number of FFP units.

web application was developed and deployed for estimating the cost differences between strategies. Additionally, the results of web application validation are reported in the Supplement.

## Clinical trial-based strategy

Transfusion protocol in neurosurgery varies across institutions; no standard guideline has been recommended for practice in neurosurgical operations from a prior systematic review [21]. For comparison, we referred to an analogous strategy from previous clinical trials [22, 23] that mentioned the number of packed red cell (PRC) and FFP units based on the FFP: PRC ratio of 1:1.

## Routine-based strategy

A routine-based strategy based on routine crossmatch at our institute was used as a reference group for comparison. A summary of preoperative blood products preparation for each

**Table 1. The strategy of preoperative blood preparation.**

| Strategy | PRC | FFP |
|---|---|---|
| Machine learning-based strategy | Number of PRC units according to web application* | Number of FFP units according to web application |
| Clinical trial-based strategy | Major operations† = 2 units | FFP preparation according to the FFP: PRC ratio = 1:1§ |
| | Endoscopic transsphenoidal approach = 1 unit | |
| | Minor operations‡ = = TS | |
| Routine-based strategy | Major operations† = 4 units | Major operation = 4 units |
| | Minor operation‡TS | |

* When ML application calculated the result as 0 unit of PRC, typing screening was used for preparation and safety for unexpected vigorous bleeding.

† Major operations were craniotomy, craniectomy, suboccipital and rectosigmoid approaches.

‡ Minor operations were burr hole with biopsy and ETV with biopsy.

§ Nascimento et al. [22] and Holcomb et al. [23]

Abbreviations: FFP = fresh frozen plasma, MSBOS = Maximum Surgical Blood Ordering Schedule, PRC = Packed red cell, TS = preoperative type and screen

strategy is shown in Table 1. Moreover, the cost per unit of blood products preparation for comparison among strategies is presented in Table 2.

## Operational definitions of the effectiveness indexes

According to the secondary objective, a comparison of the effectiveness indexes among strategies was performed; these indexes are defined as follows: [24]

C/T ratio is defined as the number of units crossmatched/number of units transfused, with a C/T ratio of 2 or less indicating effective blood utilization [25, 26].

Transfusion probability (Tp) is defined as the total number of patients transfused/total number of patients cross-matched $\times$ 100. A Tp of 30% and above indicates effective blood usage [24, 26].

Transfusion index (Ti) is defined as the number of units transfused/number of patients cross-matched. A Ti of 0.5 or more indicates effective blood usage [24–26].

## Statistical analysis

Categorical factors were presented as frequencies and percentages using descriptive statistics, whereas continuous factors were performed by mean and standard deviation.

Using data from the second cohort, the comparison of cost and effectiveness indexes among strategies was analyzed. In detail, the routine-based strategy concerned the reference group, and independent t-tests were performed to compare the mean between groups.

**Table 2. Cost of blood product preparation per unit.**

| Blood product | Cost per unit, USD* |
|---|---|
| Type and screen | 10.34 |
| Packed red cells | 47.58 |
| Fresh Frozen Plasma (1U = 250ml) | 34.72 |

* Exchange rate 1 USD = 33.77 THB. (2/12/2021)

Therefore, a p-value < 0.05 was determined as statistically significant. Statistical analysis was performed using R version 4.0.5 (The R Foundation for Statistical Computing; Vienna, Austria).

The two independent means test formula was used as well as for sample size calculation [27]. From the study of Alghamdi et al. [25], a comparison of the cost difference between a prior crossmatch strategy and a new strategy reported figures of 55,560$ and 43,316$. Moreover, we defined the acceptable standard deviation for both groups at 35,000$. Therefore, the minimum total sample size needed to test the hypothesis was 387 patients.

### Ethical considerations

The human research ethics committee of Faculty of Medicine, Prince of Songkla University approved this research (REC 64-477-10-1). Informed consent from patients was not required due to the nature of the retrospective study design. However, patients' identification numbers were encoded before analysis.

## Results

The patients' baseline characteristics and preoperative laboratories for the second cohort are presented in Table 3. The mean age was 46.36 ± 17.3 years, and the majority of patients were female (57.5%). The major underlying diseases were hypertension, dyslipidemia, and diabetes mellitus. Moreover, a preoperative seizure was observed in 10.6% of cases. Of all cases, 38.2% were meningioma, while glioma and pituitary adenomas were 30.7% and 11.4%, respectively. The mean tumor diameter was 3.13 ± 0.90, whereas the mean preoperative midline shift was 0.45 ± 0.30. By operation, more than half of all surgeries involved craniotomy with tumor removal, while the endoscopic transsphenoidal approach was performed in 12.3% of cases.

Among strategies with a 2-year time period, C/T ratio, Tp, and Ti are established in Table 4. The ML-based strategy had the lowest values for all effectiveness indexes compared to other strategies, while the routine-based strategy demonstrated a high C/T ratio and Tp that were over the reference for effective blood utilization. When comparing the indexes among strategies with quarters of the period, the mean difference of all indexes between the ML-based strategy and routine-based strategy was potentially significant, as shown in Fig 3 (p<0.001). Moreover, the mean difference of the C/T ratio and Tp between the ML-based strategy and clinical trial-based strategy was statistically significant (p 0.001 and p<0.001, respectively).

The cost and cost differences among strategies were calculated to estimate the economic impact. The overall costs of blood products preparation among the ML-based strategy, clinical trial-based strategy, and routine-based strategy were 3,0061.56$, 57,313.92$, and 136,292.94$, respectively. From the cost difference between the ML-based strategy and routine-based strategy, cost savings of 92,519.97$ (67.88%) were apparent when implementing the ML-based strategy instead of the routine-based strategy in the practice for the 2-year period. In addition, cost savings of 47.88% were observed when the ML-based strategy was implemented instead of the clinical trial-based strategy, as shown in Fig 4.

## Discussion

The results of the present study observed an imbalance between preoperative blood products preparation and actual utilization intraoperatively in brain tumor surgery over a 2-year period, in concordance with other research reports. Saringcarinkul et al. [2] studied the effectiveness indexes of 377 patients who had undergone neurosurgical operations and found that almost all brain tumor surgeries had C/T ratios over the effectiveness threshold, whereas Chotisukarat et al. [3] found C/T ratios for brain tumor operations ranged from 5–12 and Tp ranged from

**Table 3.** Baseline characteristics of the second cohort (2020–2021, N = 414).

| Characteristics | Total (%) |
|---|---|
| **Sex** | |
| Male | 176 (42.5) |
| Female | 238 (57.5) |
| **Mean age-year (SD)** | 46.36 (17.3) |
| **Age-year** | |
| 0–15 | 35 (8.5) |
| >15–60 | 299 (72.2) |
| >60 | 80 (19.3) |
| **Underlying disease** | |
| Hypertension | 78 (18.8) |
| Dyslipidemia | 62 (15.0) |
| Diabetes mellitus | 39 (9.4) |
| Liver disease | 12 (2.9) |
| Renal failure | 5 (1.2) |
| Preoperative seizure | 44 (10.6) |
| **Preoperative current medication** | |
| Antiplatelet | 9 (2.2) |
| Clexane | 2 (0.5) |
| Warfarin | 4 (1.0) |
| **American Society of Anesthesiologists classification** | |
| 1 | 1 (0.2) |
| 2 | 111 (26.8) |
| 3 | 296 (71.5) |
| 4 | 6 (1.4) |
| **Tumor classification** | |
| Meningioma | 158 (38.2) |
| Glioma | 127 (30.7) |
| Pituitary adenoma | 47 (11.4) |
| Schwannoma | 28 (6.8) |
| Metastasis | 25 (6.0) |
| Lymphoma | 4 (1.0) |
| Other | 25 (6.0) |
| **Mean diameter of tumor -cm (SD)** | 3.13 (0.9) |
| **Mean preoperative midline shift -cm (SD)** | 0.45 (0.3) |
| **Neurosurgical operation** | |
| Craniotomy | 228 (55.1) |
| Craniectomy | 39 (9.4) |
| Suboccipital or rectosigmoid approach | 53 (12.8) |
| Endoscopic transsphenoidal approach | 51 (12.3) |
| Burr hole with biopsy | 37 (8.9) |
| Endoscopic third ventriculostomy with biopsy | 6 (1.4) |
| **Emergency operation** | 65 (15.7) |
| **Mean body mass index- $kg/m^2$** | 24.00 (4.65) |
| **Mean preoperative hematocrit-%** | 38.9 (4.64) |
| **Mean preoperative hemoglobin- g/dL** | 12.84 (1.62) |
| **Mean platelet count- $x10^3/\mu L$** | 289.35 (89.71) |
| **Mean white blood cell count- $x10^3/\mu L$** | 9.88 (4.67) |

(*Continued*)

**Table 3.** (Continued)

| Characteristics | Total (%) |
|---|---|
| Mean neutrophil /lymphocyte ratio | 5.12 (7.43) |
| Mean partial thromboplastin time ratio | 0.93 (0.13) |
| Mean international normalized ratio | 1.02 (0.08) |

**Table 4. Crossmatch to transfusion ratio, transfusion probability, transfusion index of packed red cell.**

| Strategy | Number of cases | Crossmatch PRC by strategies | | Transfused PRC | | C/T ratio | Tp | Ti |
|---|---|---|---|---|---|---|---|---|
| | | Unit | Case | Unit | Case | | | |
| Machine learning-based strategy | 414 | 357 | 200 | 324 | 114 | 1.10 | 57.0 | 1.62 |
| Clinical trial-based strategy | 414 | 691 | 371 | 324 | 114 | 2.13 | 30.7 | 0.87 |
| Routine-based strategy | 414 | 1514 | 408 | 324 | 114 | 4.67 | 27.9 | 0.79 |

Abbreviations: C/T = crossmatch to transfusion, PRC = packed red cell, S = Strategy, Tp = transfusion probability, Ti = transfusion index

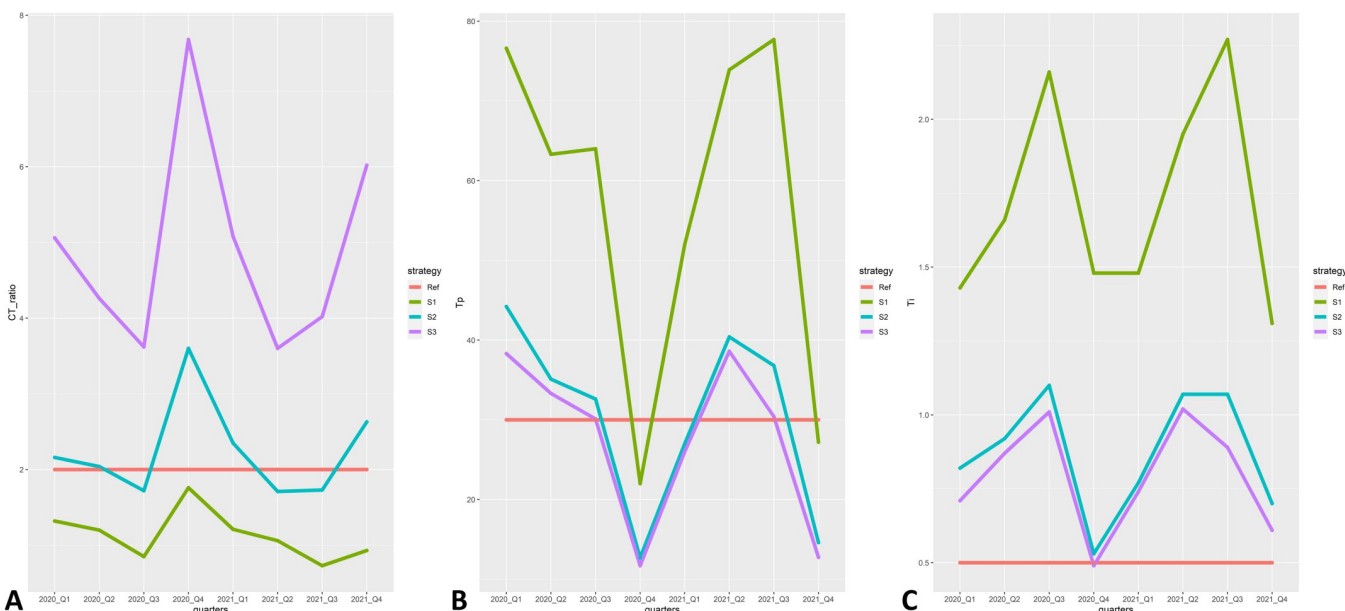

**Fig 3. Comparison of the effectiveness index for packed red cell preparation among strategies by quarters.** (A) Crossmatch to transfusion ratio, (B) transfusion probability, (C) transfusion index. Abbreviations: C/T ratio = Crossmatch to transfusion ratio, Ti = transfusion index, Tp = transfusion probability, Ref = effectiveness criteria for each index, S1 = Machine learning-based strategy, S2 = Clinical trial-based strategy, S3 = Routine-based strategy.

7–20%. In routine practice, neurosurgeons typically request more units of preoperative blood products for safety in cases of unexpected bleeding intraoperatively, leading to over-preparation. The Ti is one of the effectiveness indexes for preoperative blood preparation, but this index with a cutoff value of 0.5 may be the low threshold to detect the effectiveness of blood utilization because all strategies for preoperative blood preparation had Ti over the cutoff value. The concordant results were similar to what had been shown in previous studies [2, 3].

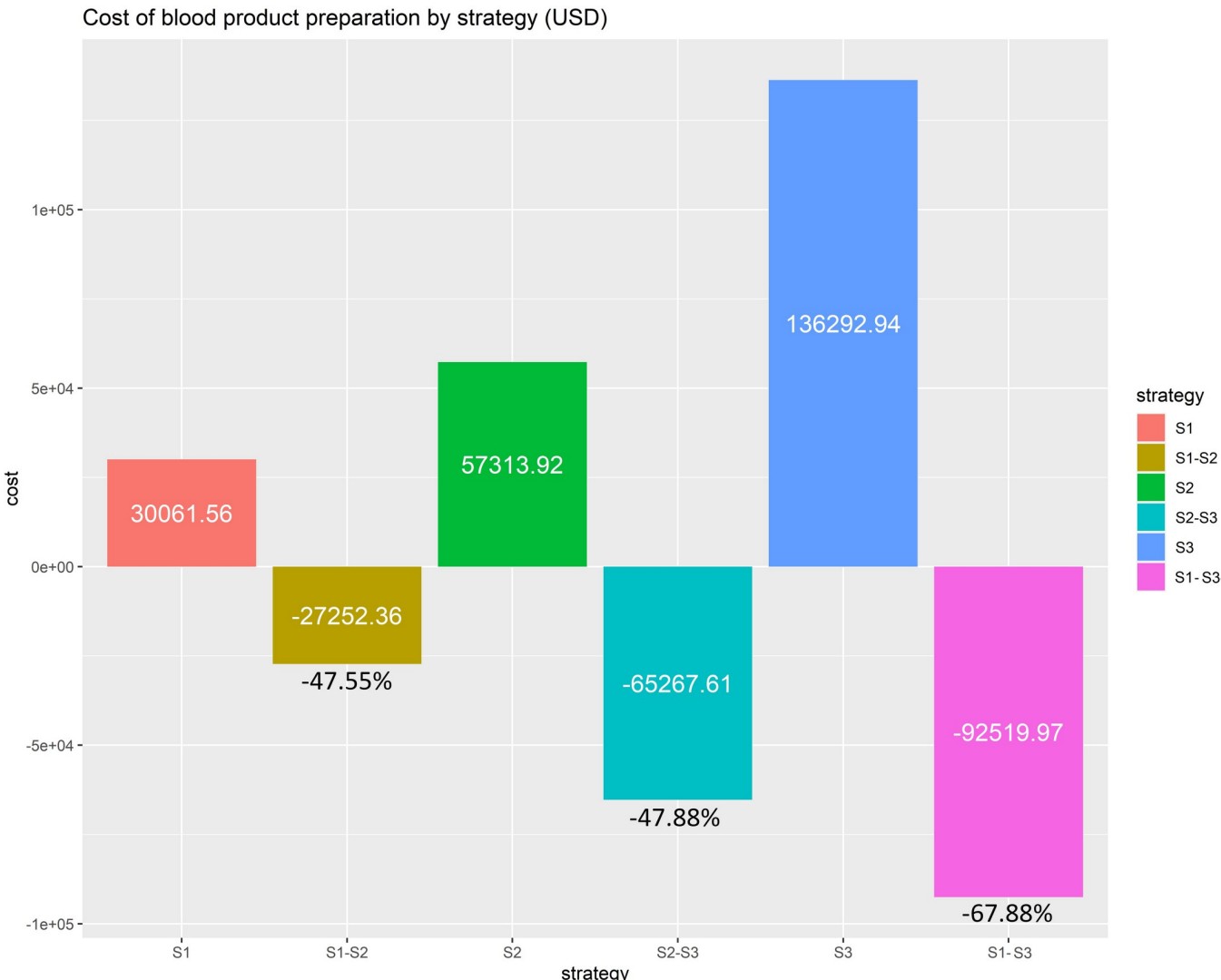

**Fig 4. Cost and cost difference of preoperative blood product preparation among strategies.** Abbreviations: S1 = Machine learning-based strategy, S2 = Clinical trial-based strategy, S3 = Routine-based strategy.

In the pandemic era, the management of limited resources should be taken into consideration since the rate of blood donation has declined globally. Several methods have been proposed to decrease the unnecessary crossmatch in the literature. Palmer et al. [28] proposed a patient-specific blood ordering system to reduce unnecessary crossmatch; this approach had sensitivity, specificity, positive predictive value, and negative predictive value of 41%, 93%, 55%, and 89%, respectively. Lam et al. [29] studied the effectiveness of a prospective physician self-audit transfusion-monitoring system for reducing unnecessary crossmatch. Although several hospitals allow the return of PRC from the ward to the blood bank if not used for transfusion, the time wasted for cross-matched preparation and unnecessary workload should be reduced. In the present study, the ML-based strategy was observed to be an effective approach to mitigate the over-requesting of preoperative blood preparation. This strategy had the lowest C/T ratio, and the highest Tp and Ti when compared with other strategies. Because the predictability of the ML-based tool exhibited an acceptable level of performance, our results

are concordant with earlier studies. Chang et al. [30] used ML with various algorithms for predicting blood transfusions in orthopedic surgery; the tool had sensitivity from 69.0–79.2%, specificity from 62.3–71.7%, and accuracy from 70.3–72.2%. Moreover, Huang et al. [31] predicted PRC transfusions in patients with pelvic fracture surgery and reported that an extreme gradient boosting algorithm enabled the best predictability with sensitivity of 93%, specificity of 97%, and accuracy of 95.1%.

In the present study, the benefits of the ML-based strategy were also assessed from the economic perspective. The significant impact of cost savings was observed when the ML-based strategy was compared to the routine-based strategy. The economic benefits were subsequently impacted by the high accuracy performance of the ML-based tool. To the best of the authors' knowledge, the present study is the first to mention the economic impact of the ML approach for MSBOS via a web application, which is one e-health platform for simplifying usage in general practice. However, the limitations of the study should also be discussed. We used local parameters for establishing costs among strategies because we needed to reflect the economic burden from the routine-based strategy in our region. Thus, generalizability is limited. Moreover, external validation of ML in terms of economic impact should be conducted at different times and places to confirm the applicability of the tool and the economic benefits as unnecessary cost savings. In addition, the present study used the same cohort to estimate cost differences among strategies. A randomized controlled trial or prospective observational study with patients assigned to different strategies may be more reflective of the real-world situation than a retrospective study design [32, 33].

## Conclusion

The ML-based strategy is an effective approach to balance the unnecessary workloads at blood banks and reduce the cost of unnecessary blood products preparation from low C/T ratio, high Tp, and high Ti. Further studies should be performed to confirm the generalizability and applicability of this strategy.

## Supporting information

**S1 File.**
(DOCX)

**S1 Data.**
(CSV)

## Author Contributions

**Conceptualization:** Thara Tunthanathip, Sakchai Sae-heng, Thakul Oearsakul, Anukoon Kaewborisutsakul, Chin Taweesomboonyat.

**Data curation:** Thara Tunthanathip.

**Formal analysis:** Thara Tunthanathip.

**Funding acquisition:** Thara Tunthanathip.

**Investigation:** Thara Tunthanathip.

**Methodology:** Thara Tunthanathip, Anukoon Kaewborisutsakul.

**Project administration:** Thara Tunthanathip.

**Resources:** Thara Tunthanathip.

**Software:** Thara Tunthanathip.

**Supervision:** Sakchai Sae-heng, Thakul Oearsakul.

**Validation:** Chin Taweesomboonyat.

**Visualization:** Thara Tunthanathip, Anukoon Kaewborisutsakul, Chin Taweesomboonyat.

**Writing – original draft:** Thara Tunthanathip.

**Writing – review & editing:** Sakchai Sae-heng, Thakul Oearsakul, Anukoon Kaewborisutsakul, Chin Taweesomboonyat.

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
