## [Decision Letter · Decision Letter 0]

4 Apr 2022

PONE-D-22-00205Economic Impact of a Machine Learning-Based Strategy for Preparation of Blood Products in Brain Tumor SurgeryPLOS ONE

Dear Dr. Tunthanathip,

Thank you for submitting your manuscript to PLOS ONE. After careful consideration, we feel that it has merit but does not fully meet PLOS ONE’s publication criteria as it currently stands. Therefore, we invite you to submit a revised version of the manuscript that addresses the points raised during the review process.

ACADEMIC EDITOR:Please revise the manuscript in line with the reviewers' comments.

We look forward to receiving your revised manuscript.

Kind regards,

Venkatesh Shankar Madhugiri

Academic Editor

PLOS ONE

Journal Requirements:

2. Please include your tables as part of your main manuscript and remove the individual files. Please note that supplementary tables (should remain/ be uploaded) as separate "supporting information" files.’

“The Faculty of Medicine, Prince of Songkla University. (Grant no. 64-477-10-1)”

Additional Editor Comments (if provided):

Please see reviewers comments and address the issues raised by them.

Reviewers' comments:

Reviewer's Responses to Questions

**Comments to the Author**

1. Is the manuscript technically sound, and do the data support the conclusions?

Reviewer #1: Partly

Reviewer #2: Yes

2. Has the statistical analysis been performed appropriately and rigorously? 

Reviewer #1: Yes

Reviewer #2: Yes

3. Have the authors made all data underlying the findings in their manuscript fully available?

Reviewer #1: Yes

Reviewer #2: Yes

4. Is the manuscript presented in an intelligible fashion and written in standard English?

Reviewer #1: Yes

Reviewer #2: Yes

5. Review Comments to the Author

Reviewer #1: Economic Impact of a Machine Learning-Based Strategy for Preparation of Blood Products in Brain Tumor Surgery

The main objective of this study was to identify the cost differences of the ML-based strategy compared with other strategies in preoperative blood products preparation.

For the reader it is hardly possible to compare the strategies. No lab parameters could be found as well as no evidence-based strategies. Therefore, the different strategies must be discussed in more detail.

In some university hospitals it is allowed to take back packed red blood cells (pRBC) from the ward to the blood bank (established quality management system). This is not allowed in the US, but in Europe. Maybe you can talk with the responsible persons in your hospital. This must not be considered here.

ML is the strategy of choice and should be introduced wherever it is possible. Does it work also in catastrophes (Blackout)? Perhaps you can prepare another paper to answer this question.

Here some hints:

Introduction

References 2, 3 of blood transfusion in neurosurgery in Thailand with limited meaningfulness. See for example:

Curr Opin Anaesthesiol. 2014 Oct; 27(5): 470–473. doi: 10.1097/ACO.0000000000000109

McGirr, A, Pavenski, K, Sharma, B, Cusimano, M. Blood conservation in neurosurgery: erythropoietin and autologous donation. Can J Neurol Sci. 2014;41(5):583-589.

Meybohm, P., Fischer, D.P., Geisen, C. et al. Safety and effectiveness of a Patient Blood Management (PBM) program in surgical patients - the study design for a multi-centre prospective epidemiologic non-inferiority trial. BMC Health Serv Res 14, 576 (2014). https://doi.org/10.1186/s12913-014-0576-3

Goebel, BJA Patient blood management in intracranial neurosurgery—do we have sufficient data to define a transfusion threshold? Comment on Br J Anaesth 2018; 120: 988-98

No reference is given although it is used as supplementary data

Please, make an update of the literature

Guidelines

Reference 18, 19 are no guidelines, but clinical trials.

Clinical Neurology and Neurosurgery, Volume 155, April 2017, Pages 83-89: Evidence-based outcomes for transfusion thresholds and indications are limited.

Please, discuss the advantages and limits of your references in respect of missing guidelines.

Costs

$3,0061.56, $57,313.92, and $136,292.94

Please, correct the error and use US-$. How did you calculate the costs? Since the prices of packed red blood cells are very different in different countries, I recommend giving percentages with reference to one of the basic methods.

Conclusions are too general

The pandemic has caused a reduction in blood donation activity. Maybe this is true in Thailand, but not in other countries due to different rules to live with the pandemics.

This general conclusion is not valid. It is an assumption, but not shown with data. It seems that partly the opposite is the case.

ML-based strategy offers a way to calculate preoperative crossmatch and effectively reduce the cost of unnecessary blood products preparation via a web application.

Cross match does not mean that the pRBC are on the ward and either transfused or lost if not needed. As long as the pRBC are in the refrigerater on ward or stil in the blood depot of the blood bank the pRBC can be used also for other patients.

Please, formulate more precisely.

Reviewer #2: In these retrospective study the authors aimed to analyzed patient who received surgery from January 2014 and December 2021. Through the analysis of multiple preoperative parameters that may affect the RBC transfusion volume, they used ML algorithms to build up the artificial intelligence (AI) model to predict the accurate RBC demand quantity and compared each result with those predicted by clinicians and to evaluate the economic impact. Their results were shown that the ML methods is more accurate than clinician experience in predicting preoperative RBC transfusion, which reduce the unnecessary cost of blood preparation.

This seems a promising study, can be accepted

6. PLOS authors have the option to publish the peer review history of their article (what does this mean?). If published, this will include your full peer review and any attached files.

Reviewer #1: No

Reviewer #2: No

---

## [Author Response · Author response to Decision Letter 0]

11 Apr 2022

I revised the manuscript as the reviewer’s suggestion. (response to reviewer file)

---

## [Decision Letter · Decision Letter 1]

20 Jun 2022

Economic Impact of a Machine Learning-Based Strategy for Preparation of Blood Products in Brain Tumor Surgery

PONE-D-22-00205R1

Dear Dr. Tunthanathip

We are pleased to inform you that your manuscript has been judged scientifically suitable for publication and will be formally accepted for publication once it meets all outstanding technical requirements.

Within one week, you will receive an e-mail detailing the required amendments. When these have been addressed, you will receive a formal acceptance letter and your manuscript will be scheduled for publication.

Kind regards,

Venkatesh Shankar Madhugiri

Academic Editor

PLOS ONE

---

## [Editor Report · Acceptance letter]

24 Jun 2022

PONE-D-22-00205R1 

Economic Impact of a Machine Learning-Based Strategy for Preparation of Blood Products in Brain Tumor Surgery 

Dear Dr. Tunthanathip:

I'm pleased to inform you that your manuscript has been deemed suitable for publication in PLOS ONE. Congratulations! Your manuscript is now with our production department. 

Kind regards, 

on behalf of

Dr. Venkatesh Shankar Madhugiri 

Academic Editor

PLOS ONE